# No Worm Is an Island; The Influence of Commensal Gut Microbiota on Cyathostomin Infections

**DOI:** 10.3390/ani10122309

**Published:** 2020-12-05

**Authors:** Nicola Walshe, Grace Mulcahy, Jane Hodgkinson, Laura Peachey

**Affiliations:** 1School of Veterinary Medicine, University College Dublin, 4 Dublin, Ireland; grace.mulcahy@ucd.ie; 2Institute of Infection and Global Health, University of Liverpool, Liverpool L69 7ZJ, UK; jhodgkin@liv.ac.uk; 3Bristol Veterinary School, Faculty of Health Sciences, University of Bristol, Langford BS40 5DU, UK

**Keywords:** microbiome, equine, helminth, cyathostomin, cyathostominosis, immunity

## Abstract

**Simple Summary:**

There is increasing evidence for the importance of gut bacteria in animal health and disease. This is particularly relevant for gastrointestinal infections, such as parasitic worms, which share a niche with gut bacteria. Parasitic worms are highly prevalent in domestic horses and are a significant cause of disease in this population. This commentary explores the complex relationships between the most common parasitic worm in horses (cyathostomins) and gut bacteria, based on recent studies in horses and other species. We propose novel theories and avenues for research that harness these relationships and have the potential to improve control of parasitic worms, and overall equine health, in the future.

**Abstract:**

The importance of the gut microbiome for host health has been the subject of intense research over the last decade. In particular, there is overwhelming evidence for the influence of resident microbiota on gut mucosal and systemic immunity; with significant implications for the outcome of gastrointestinal (GI) infections, such as parasitic helminths. The horse is a species that relies heavily on its gut microbiota for GI and overall health, and disturbances in this complex ecosystem are often associated with life-threatening disease. In turn, nearly all horses harbour parasitic helminths from a young age, the most prevalent of which are the small strongyles, or cyathostomins. Research describing the relationship between gut microbiota and cyathostomin infection is in its infancy, however, to date there is evidence of meaningful interactions between these two groups of organisms which not only influence the outcome of cyathostomin infection but have long term consequences for equine host health. Here, we describe these interactions alongside supportive evidence from other species and suggest novel theories and avenues for research which have the potential to revolutionize our approach to cyathostomin prevention and control in the future.

## 1. Introduction

The importance of the human gut microbiome in health and disease has achieved significant prominence over the last decade, in parallel with the technological advances that have enabled cost-effective and accurate metagenomic analysis. Microbiome modulatory treatments, such as faecal microbial transplants (FMTs), are now being used to treat certain conditions in humans, e.g., *Clostridium difficile* infection [1,2]; and further treatments are in the pipeline. Whereas rodent studies have been used to model some aspects of gut dysbiosis and their importance to human health, similar studies in veterinary species, such as the horse, have lagged behind. Although physiologically distinct from humans, horses suffer from many analogous diseases in which the gut microbiome plays a role; for example, inflammatory bowel disease (IBD) [3,4], equine metabolic disease (EMS) (akin to type 2 diabetes) [5], recurrent airway obstruction (RAO) (akin to asthma) [6] and gastrointestinal (GI) helminth infections [7,8,9].

The relationship between the gut microbiota and parasitic helminths has received significant attention in humans; in particular, regarding the role of the microbiota in modulating host immune responses to helminth infection, and the knock-on consequences for the long-term health of the host [10]. The loss of the immune-regulation associated with helminth infection may, directly or indirectly, contribute to the high rate of human autoimmune and allergic disease in the developed world, where helminth infection is not prevalent; described as the “hygiene hypothesis” or “old friends” hypothesis [10,11]. In fact, helminth therapy has been proposed as a therapeutic tool for Th1 driven autoimmune diseases in humans [10,12,13]; and, for example, has been shown to reduce the severity of colitis in animal models [14,15]. All horses with access to grazing have a resident population of parasitic helminths comprising mostly of mixed infections with the cyathostomins (small strongyles) [16], of which over 50 species have been documented [17]. Cyathostomins have the potential to be highly pathogenic [9,16], although many horses harbour significant burdens with no adverse effects. Consequently, in the domesticated horse, cyathostomin infections are routinely suppressed from a young age using an anthelmintic treatment. Frequent use of anthelmintics has resulted in the emergence of widespread anthelmintic resistance in cyathostomin populations globally, which is an increasing threat to the sustainable control of these parasites. The impact of this approach on overall gut health has not been studied systemically. A handful of studies to date have described interactions between the gut microbiota and cyathostomins [7,8,9,18,19], and the results suggest a complex inter-relationship between these two groups of organisms.

At present, significant limitations prevent our full understanding of the relationship between the equine microbiota and chronic cyathostomin infection. Firstly, to date, systematic studies on the equine gut microbiome are limited. At the time of writing, a search of PUBMED using the terms “equine” and “microbiome” yields a total of 234 papers; for “equine” and “helminth” and “microbiome” the results are far fewer (*n* = 4). Most of the studies which have been carried out to date involve small numbers of animals and use 16sRNA analysis, and thus lack power and functional inference. Furthermore, a recent meta-analysis from our group has revealed that technical factors such as gene region and sequencing platform have an overwhelming influence on sequencing data in comparison to biological factors [20]. Nonetheless, there is consensus between studies with regards to the core equine faecal microbiome. O’Donnell et al. [21], for example, reported the dominance of the phyla Firmicutes, Bacteroidetes, Proteobacteria, Verrucomicrobia, Actinobacteria, Euryarchaeota, Fibrobacteres and Spirochaetes in six thoroughbred racehorses in Ireland; many subsequent studies have reported similar overall abundances. For further detail, Costa and Weese [22] provide a concise review of recent studies on the equine faecal microbiota.

In order to accelerate progress in this complex field, this review presents a synthesis of what is currently known about the gut microbiota in equids, in conjunction with how the gut microbiome might be influenced by, and interact with, resident cyathostomin populations. We propose that resident cyathostomin populations be considered as part of the gut flora, along with the microbiota, and that they play an important role in the systems biology of the equid gut. We suggest how the understanding, treatment and control of cyathostomin infection can be advanced through improved characterisation of the equid gut flora and highlight some important research gaps and pathways for investigation. Finally, we propose that the understanding of acute and chronic inflammatory states in horses, in which gut microbiota are implicated, may improve approaches to managing overall equine health in future.

## 2. Evidence for Cyathostomin–Microbiome Interactions in Horses

There is an increasing body of evidence reporting complex inter-relationships between cyathostomin infections and the gut microbiota in horses. These can be broadly broken down into two-way interactions between the host-microbiota, helminth-host, and helminth microbiota (see Figure 1); however, these relationships are often bidirectional and involve complex, as yet poorly understood, feedback loops.

### 2.1. Early Life Programming of Host Immunity by Microbiome

Perhaps the most important time period to evaluate these relationships is during the neonatal period when the gut microbiome and immune system of vertebrates are malleable and undergoing “programming” [23,24]. Data on the developing gut microbiota in foals show that the transition to an adult microbiome involves a rapid increase in microbial diversity, and an increasing proportion of fibre digesting taxa such as Fibrobacteres [25,26,27,28]; as horses are hind-gut fermenters, this process is key to the utilisation of dietary fibre and optimisation of gut health. Foals naturally ingest the dam’s fresh faeces between two to five weeks of age [29], and it is during this time that the most rapid increase in diversity and change in the composition of the faecal microbiota occurs [26,28]; suggesting that GI tract colonisation may be aided by coprophagia. Based on data from other mammals, we also know that the process of gut colonisation has a long-lasting impact on immune function [23]. Specifically, a reduction in microbial alpha-diversity in the first week of life, for example, due to caesarean section, is associated with the development of atopy and allergies in humans [30,31,32,33]. Little is known about the impact of microbiome development on long-term immunity in horses, but it stands to reason that similar interactions occur between the gut microbiota and immune-homeostasis in all vertebrates.

Consequently, a key question regarding cyathostomin–microbiome interactions in horses, is whether the composition of the early-life gut microbiome affects the host immune response to cyathostomins. Foals in domesticated environments are generally subject to different environments than those in the wild; for example, the external environment and mares’ perineal area will be cleaner, and plasma and antibiotics are commonly administered in the first few days of life [34,35]. Given that foals exhibit coprophagia there may also be differences between the transgenerational transfer of microbiota from wild versus domesticated dams. These factors will all affect the colonisation of the GI tract during this key period. Notably, a study in adult Prejwalski’s horses showed that a subgroup of a wild herd which had been born in captivity had significantly lower microbial alpha-diversity than their wild-born counterparts [36]. As mentioned above, a reduction in diversity in early-life is a risk factor for disruption of immune homeostasis; therefore, birth in captivity may, in theory, represent a risk factor for altered immunity to helminths. In support of this hypothesis, a study in frogs showed that antibiotic use in juveniles caused a reduction in microbial alpha-diversity and concomitant increase in helminth burdens in adults [37]. Furthermore, in rodents, supplementation with a specific microbe (*Lactobacillus taiwanensis*) prior to experimental helminth infection resulted in increased helminth burdens [38]. These data support the theory that the composition of the microbiota, particularly in early-life, can influence the outcome of helminth infection later in life. No study, to date, has tested this hypothesis in horses, and we suggest this is an important area for future research.

### 2.2. Impact of Acute Cyathostomin Infection on the Gut Microbiome

Approaching cyathostomin–microbiome interactions from another angle, it is equally possible that helminth infections themselves cause fluctuations in developing microbiota, which subsequently has a long-term effect on equine health. In particular, young animals reared in highly-stocked paddocks will be subject to high infection challenge from cyathostomins. Previous studies have shown that acute cyathostomin infection in young animals causes significant disruption to the gut microbiota [7,8,9,18], which was reversible upon treatment in one study [8]. These changes varied according to study but included evidence of dysbiosis, including a reduction in the microbial richness of the faecal microbiota; increases in the abundance of taxa with potential immune-modulatory functions, such as *Lactobacillaceae* (Bacilli), *Mogibacteriaceae* (Clostridia); increases in pathobionts such as *Campylobacter* and *Pasteurella* (Proteobacteria); and reductions in taxa associated with health such as *Lachnospiracaeae* (Clostridia). As discussed above, a reduction in diversity, alongside these specific compositional changes could potentially have impacts on long-term GI/immune homeostasis in young horses. The net effect of these changes, and investigations into their long-term impacts on equine health, should be the subject of further research.

### 2.3. The Relationship between the Gut Microbiome and Immunoregulation in Chronic Cyathostomin Infection

The response of the gut microbiota to chronic cyathostomin infection seems to be distinct from that observed in acutely-infected horses. Data from adult horses show very little perturbation due to chronic moderate infection, which tended to increase microbial richness [7]. This has also been observed in previous studies in humans, where increased microbial diversity was observed in people chronically infected with both *Strongyloides stercoralis*, and mixed helminth infections [39,40]. It is well established that helminths in general have a suppressive effect on Th1 mediated immune responses, and horses are no exception. Although immunological studies of cyathostomin infection are few, reports to date have described a Th2/Treg type response to infection in the GI mucosa of chronically infected horses [41,42]. Furthermore, evidence from rodents demonstrates that some of these effects can be mediated by alterations in the gut microbiota [43,44]. As discussed above, removal of helminths is linked with an increased prevalence of autoimmune disease in humans. Furthermore, horses also suffer from a number of immune-mediated diseases which cause significant morbidity and mortality to the global equine population; these include IBD, atopy and RAO. It is plausible that chronic suppression of cyathostomin infection could increase the risk of developing some of these immune-mediated diseases. Indeed, two studies showed higher FECs in healthy controls when compared to horses with RAO [45,46]; suggesting a possible link between cyathostomin infection and the prevention of allergic disease. Therefore, we hypothesise that helminth-induced moderations of the gut microbiota could be beneficial in the prevention, and possibly treatment, of allergic/autoimmune diseases in horses. Further research in this field will not only advance our understanding of equine health but will have important implications for similar disease in other species, including humans.

In summary, it is certain that cyathostomins induce changes in the gut microbiota which alter the complex cross-talk occurring between microbes and the mucosal immune barrier, but further research is needed to evaluate the causality of these interactions, and short and long-term consequences for equine health. As with any complex symbiotic system, it is likely that a balance exists whereby acute, heavy infection will have immediate detrimental effects on the microbiome with possible long-term implications; whereas moderate, chronic infection is likely to be associated with health benefits, some of which are mediated by the gut microbiome. The challenge now is to unravel these complex effects to identify how microbiome modulation may play a role in the prevention and management of cyathostomin infection and overall equine health.

## 3. The Role of Equine Gut Microbiota in Acute Larval Cyathostominosis

Whilst significant burdens of cyathostomins can be tolerated by both immature and adult horses, under certain conditions cyathostomins can lead to acute life-threatening disease, known as acute larval cyathostominosis (ALC). Understanding interactions between the host and microbiota in this condition may influence how this disease is prevented and treated and thus deserves in-depth focus. The potential role of the gut microbiota in this condition can broadly be split into two areas: (1) the initiation of ALC and (2) the pathophysiology of ALC (Figure 2).

### 3.1. The Role of the Gut Microbiota in the Initiation of ALC

Cyathostomins have a direct life cycle with infection through the faecal–oral route. The predilection site for these endoparasites is the caecum and large colon, where they encyst within the mucosa. Within the mucosa the larvae can become inhibited through hypobiosis and can remain in the large intestinal walls for up to three years or more [47], thereby, enabling large burdens to accumulate in certain conditions. ALC is attributed to the emergence “en masse” of previously encysted (possibly hypobiotic) larvae from the mucosa from the large intestine wall resulting in a severe typhlocolitis. The most common presenting signs include severe weight loss, diarrhoea and pyrexia [48,49], with weight loss sometimes being the only or at least the initial presenting sign [50]. Cases have developed endotoxaemia purported due to intestinal wall compromise [9,51]. A high mortality rate has been reported, with up to 50% mortality in intensively managed cases in referral hospitals [48,49].

Risk factors associated with the disease include time of year, age and previous anthelmintic treatment [52]. Encysted cyathostomin burdens are likely to be augmented by successive waves of ingested larvae becoming hypobiotic; which is proposed to be due to population pressures and environmental conditioning of the larvae [53,54] and infectious burden on pasture. The majority of reported cases occur in winter and spring [9,50,55,56], there is speculation that this due to an emergence of larvae in response to natural expulsion of adult worms with decreased fecundity as part of yearly cycles [54,57,58]. Removal of adults through anthelmintic treatment has also been associated with the development of ALC [9,16]. ALC is commonly associated with younger animals, and this is supported by several reports on both outbreaks and hospital reports [9,49,50]. However, there are often horses who develop ALC without the above risk factors [59] and outbreaks that only affect certain cohorts of the herd [9,60]. This has led to speculation that other, not yet known, confounding factors may be at play regarding individual susceptibility to and varying presentation of ALC.

As discussed above, there is growing evidence of a tolerogenic relationship between helminths, the gut microbiota and the host immune response in the intestinal ecosystem [14,38,43,61]. Furthermore, immune modulation by helminths may be partially mediated through the manipulation of the intestinal microbiota [43,44,62]. Interference with this steady-state of helminth-microbiota–host interaction could lead to disruption of delicate balance and removal of the “immunoregulatory brake”. For example, in the case of ALC occurring post-adulticide treatment, the removal of luminal worms might trigger changes to microbiota which alter the host Th1/Th2/Treg balance and lead to recognition of encysted larvae. Finding such a trigger may, in theory, offer novel approaches to the prevention of this disease in future.

### 3.2. The Role of Gut Microbiota in the Pathophysiology of ALC

The gut microbiota are also strongly implicated in exacerbating the severity of mucosal inflammation once the mass emergence of larvae has begun. A recent case series demonstrated extensive invasion of damaged mucosa by bacteria at post-mortem [9]. Furthermore, the most severely affected cases suffered from systemic endotoxemia due to bacterial translocation through the mucosa [9]. In these cases, ALC was associated with significant disruption of faecal microbiota, with a reduction in richness and increases in bacteria such as *Lactobacillaceae* and *Streptococcus*. In rodent models of *Schistosoma mansoni* infection, a similar role for the microbiota has been described. *S. mansoni* is a trematode which infects humans in tropical and sub-tropical regions; like cyathostomins, part of its life cycle relies on the migration of immature stages (in this case eggs) across the GI mucosa. A study by Jenkins and colleagues [63] demonstrated that this process was accompanied by significant dysbiosis, including a reduction in richness and increases in *Lactobacillaceae*. A further study showed that antibiotic administration during infection significantly reduced pathology and limited the passage of eggs across the gut wall [64]. This is strong evidence that bacteria may not only increase pathology in acute *S. mansoni* infection but may also facilitate the transit of eggs across mucosa by promoting inflammation. It is highly plausible that similar mechanisms occur in other helminth infections, such as ALC.

Thus, with the presentation of more severe cases with endotoxaemia, treatment regimens should keep the tripartite relationship between microbiota, host and cyathostomins in mind. Several aspects should be considered; the emerging larvae, the unregulated immune response, the possible involvement of disruption of the gut microbiota and the loss of function of the hindgut fermentation process. Specifically, the consideration of antibiotic administration should be made on a case by case basis, in line with signs of a compromised gut wall.

## 4. Future Therapeutic Options Based on Microbiota Modulation

As we have outlined in Section 1, Section 2 and Section 3 of this article there are still many unknowns regarding the impact of the gut microbiota on immunity to cyathostomins, and the impact of cyathostomins on the microbiota (and consequently health) of the horse. As such, it is not possible to recommend specific microbiome-modulatory approaches to aiding cyathostomin control at this point in time. However, we can suggest directions for future work to generate progress in this area.

### 4.1. Microbiota Modulating Tools

Currently, the main tools used to modulate gut microbiota are probiotics, prebiotics and faecal microbial transplants; although more refined tools to target specific bacteria in the gut (e.g., use of bacteriophages) may become available in the coming years. Theoretical ways in which these treatments might affect cyathostomin infection outcome are represented in Figure 3.

Probiotics are live microorganisms that are administered orally for purported health benefits. Historically, the evidence of their efficacy to treat specific diseases in any species is weak. Probiotics used in the context of equine disease, (reviewed by Schoster [65,66]), revealed conflicting evidence, with indications for their use being insufficient. The most likely reasons for the failure of probiotics to improve disease outcomes in horses to date is that the taxa included in formulations are chosen empirically based on data from other species with the most common bacterial probiotics being *Lactobacillus*, *Bifidobacterium* and *Enterococci* [65] which are present in very low abundance in the large colon of the horse [67,68,69]. Furthermore, the survival of bacteria during passage through low pH stomach fluid and the digestive process may limit the dose reaching the target site with short faecal survival times seen in both bacterial and yeast probiotics studies [70,71]. Prebiotics are products that are intended to alter microbiota by acting as selective substrates for host microbiota. Some of the more common include oligofructose, inulin, fructooligosaccharides and mannan-oligosaccharides. Again, the evidence of their effects on gut microbiota is conflicting, however, some positive benefits have been shown in their use to mitigate microbial disruption with abrupt changes of diet [72]. Faecal microbiota transplants (FMT) have been shown to be very successful in the treatment of *Clostridium difficile* infections in humans [1,73] and with variable results in the treatment of IBD, but some promise with regard to ulcerative colitis [74]. The use of FMTs in horses is mostly anecdotal with minimal research in the area. However, there are some promising indications of efficacy, in that FMTs have been shown to increase the alpha diversity of geriatric horses with colitis [75] and had significant rapid results in horses treated for acute onset colitis in hospitalised horses post colic surgery [76]. FMT may have a role in restoring gut function in horses suffering from dysbiosis. However, NG administration would be problematic due to the possible degradation of the bacteria at the level of the stomach, small intestine and caecum [77]. Therefore, more research is needed to establish best practice guidelines for the following criteria, as recommended by Mullen [77]: transplant material type, donor characteristics, donor screening, transplant method and disease target.

### 4.2. Scope for Prevention of Cyathostomin Infection

Firstly, given the evidence that early life microbiota may influence susceptibility to helminth infection in other species, research should focus on microbiota modulatory approaches to enhancing a robust response to helminth infection in early life. Such approaches might include administration of pre/probiotics, FMTs (e.g., from horses with proven resistance), or encouraging the natural transfer of microbiota from the dam/environment to the foal and minimising interventions which may interfere with normal colonisation of the GI tract. Key areas for future research should include identifying which features of early life gut microbiota are important for immune development. In addition to observational studies, which are often confounded by variation, controlled experiments in which early life GI colonisation is altered, followed by experimental infection, are essential to advance knowledge in this area.

In relation to the prevention of life threatening ALC, it is possible that changes in the gut microbiota are involved in the triggering mechanisms for mass emergence of larvae. Therefore, future research in this area could include monitoring high-risk animals during outbreaks for changes to the microbiome which precede larval emergence. Furthermore, once larval emergence has been triggered, the inflammation resulting from the secondary invasion of bacteria may facilitate and promote further larval emergence. Therefore, future treatment trials should address whether early antibiotic use could improve outcomes.

### 4.3. Scope for Treatment of Cyathostomin Infection

It may also be possible to use microbiota-modulatory treatments to treat the cyathostomin infection. For example, a prebiotic fructan, inulin, has been shown to exert significant anthelmintic activity against GI nematodes in pigs [78,79]. The mode of action is through a reduction in pH at the site of infection, due to the proliferation of lactic acid-producing bacteria which use inulin as a substrate [80]. This particular approach would not be suitable for horses as the resultant large intestinal acidosis could be fatal. However, in the future, other modifications may be found which have anthelmintic effects without posing a health risk. For example, in rodent models, the GI helminth *Trichuris muris* has been shown to induce modifications to the host microbiota which reduces the success of subsequent helminth infection [81]; describing these changes could prove key to informing the design of future treatments to prevent *T. muris* infections. The same relationships may also exist in equine cyathostomin infection. We currently have little or no relevant mechanistic data on this topic, however, primarily because horses are a far less tractable and economical system for this type of research. If we are to create possibilities for improving equine health through microbiome manipulation, this is the type of research that is needed.

The rationale for antibiotic use in some cases of ALC has been discussed above. Given the severe dysbiosis and loss in gut microbial diversity that has been reported in this condition [9], future treatments could also involve the repopulation of the hindgut by using fibre-based feeding or even FMT, provided that an evidence base for such interventions could be established.

### 4.4. Considering the Microbiota and Overall Gut Health in Long-Term Management of Cyathostomins

Finally, we should consider how we can manage cyathostomin infections to minimise any long-term deleterious effects of cyathostomin infection on the equine gut microbiota. Changes in management and the use of ML anthelmintics have both been identified as risk factors for the development of colic and ALC [9,82]. Evidently, the stability of management, and hence the microbiome, is vitally important for equine gut health. We know that acute cyathostomin infection causes significant shifts in gut microbiome composition and diversity and that anthelmintic treatment of these cases further alters the microbiota [8,9,19]. Indeed, a recent study from our group demonstrated concomitantly greater variation in microbial diversity and parasite burden in ponies subject to poor pasture management and sporadic anthelmintic treatment [83]. Thus, we propose that keeping horses on pasture with high infection levels, followed by repeated use of an anthelmintic, is likely to cause significant fluctuations in the composition and diversity of gut microbiota, which may, in turn, result in poor gut health. Emphasis should be on limiting infection challenge, minimal use of drugs, and gradually allowing a degree of natural immunity to develop. In turn, this would aid the development of low-moderate chronic infection which may have long-term health benefits. Fortunately, this is exactly the approach which is advocated to reduce selection pressure for anthelmintic resistance and hopefully will provide further incentive to owners and managers to prioritise managemental approaches to cyathostomin control, for the overall health of their horses, in addition to the sustainable use of anthelmintics.

## 5. Conclusions

The tripartite relationship between the cyathostomin, gut microbiota and host immune response is complex, with implications for prevention and exacerbation of disease states. A deeper understanding of the mechanisms involved is crucial for harnessing these interactions to improve equine health. In particular, the interaction between helminths, the gut microbiota and host physiology in early life may be a key target for future interventions to improve cyathostomin control. Future research should focus on understanding how the development of microbiota in early life influences immunity to helminths, how heavy infections in early life influence long term health, and finally, how the gut microbiota may be responsible for initiating or exacerbating ALC.

## Figures and Tables

**Figure 1 animals-10-02309-f001:**
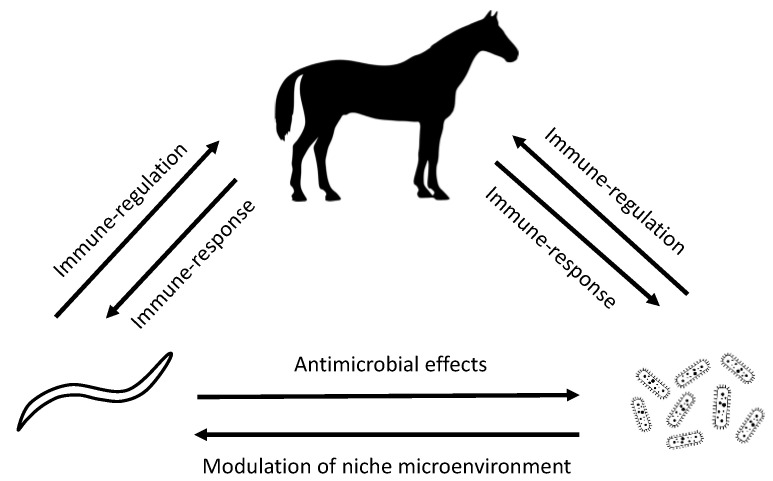
Diagrammatic representation of the theoretical interrelationships between horses, microbiota and cyathostomins.

**Figure 2 animals-10-02309-f002:**
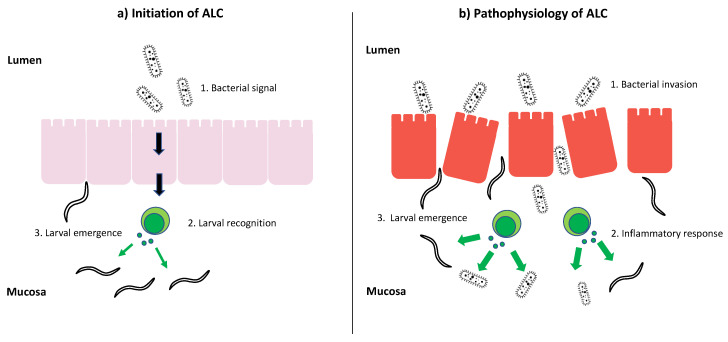
Diagrammatic representation of a theoretical role for gut microbiota in (**a**) the initiation of ALC; where a change in the microbiota may signal to mucosal immune cells to recognise larvae, and (**b**) the pathophysiology of ALC; where the invasion of damaged mucosa by gut microbiota may exacerbate inflammation, cause further activation of larvae and facilitate their passage to the gut lumen.

**Figure 3 animals-10-02309-f003:**
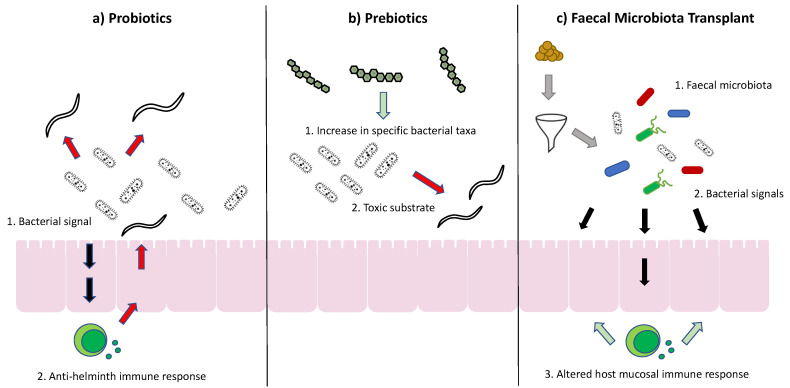
Diagrammatic representation of the suggested mode of action of (**a**) probiotics; where specific taxa may alter the host immune response to cyathostomins, (**b**) prebiotics; which may cause an increase in certain taxa which produce substrates which kill cyathostomins, and (**c**) faecal microbial transplants; which may introduce taxa which alter host mucosal immune defences.

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
