# Peer review of "No Worm Is an Island; The Influence of Commensal Gut Microbiota on Cyathostomin Infections"

_animals, 2020, doi:10.3390/ani10122309_

Round 1

Reviewer 1 Report

In this commentary, the authors have explored the complex relationships between the most common parasitic worm in horses (cyathostomins) and gut bacteria, based on recent studies in horses and other species.

The paper is well written and I do not have any major comments/concerns. 

The authors have suggested/proposed various ways of prevention and treatment of cyathostomin infections. However, it would be good to also include a section outlining the current measures that are employed in the prevention and treatment of cyathostomin infections, if any, and why those measures haven't been sufficient.

Author Response

Response to reviewer 1:

We thank reviewer 1 for their positive comments regarding our article.

We appreciate that outlining current measures for prevention would add another dimension to our article; however, the focus of this review is on the relationship between gut bacteria and cyathostomins, and we do not feel it would add to the overall message to go into current control measures in detail. We have added another line 63-65, to highlight the threat of anthelmintic resistance; and furthermore, have touched on this topic throughout the article when relevant.

Reviewer 2 Report

This is a well-researched and thorough review of an important topic that has received insufficient attention to date.  The authors have identified what is known and, perhaps more importantly, where the current gaps in our knowledge exist.  The information that is presented is free of bias and provides the reader with sufficient information to assess the importance of this topic.  I appreciated their attention to detail and the thoroughness of their efforts.

Author Response

We thank reviewer 2 for their positive comments, and hope that they enjoyed reading this article.

Reviewer 3 Report

This review summarizes the knowledge on the effect of the gut microbiome on parasite infection, in particular cyatostomin infections in horses. This is for sure an up to date topic and I think it is important to promote these ideas that will be relevant for other host parasite systems as well. I found the article interesting and well written. I think it includes all major aspects and suggestions for future research. In my opinion this article is ready to be published almost as is.

Only some small suggestions:

L30: “…control in the future.”

L112: “.” instead of “?”

My main concern is about the figures. It would be nice if you could make the “worm” look more like a nematode instead of an earthworm. This should be possible without much effort.

Maybe it is possible to make Fig. 2 and 3 a bit more self-explanatory by including some explanation in the figure itself (at the moment, the figures are only understandable when reading the caption).

Author Response

We thank reviewer 3 for their positive comments about our article.

We have made the minor amendments and suggested adjustments to figures that they suggested.